# Seeing is Not Necessarily Believing: Limitations of BigGANs for Data Augmentation

**Suman Ravuri, & Oriol Vinyals**
DeepMind
London, UK
`{ravuris,vinyals}@google.com`

## Abstract

Recent advances in Generative Adversarial Networks (GANs) – in architectural design, training strategies, and empirical tricks – have led nearly photorealistic samples on large-scale datasets such as ImageNet. In fact, for one model in particular, BigGAN, metrics such as Inception Score or Frechet Inception Distance nearly match those of the dataset, suggesting that these models are close to matching the distribution of the training set. Given the quality of these models, it is worth understanding to what extent these samples can be used for *data augmentation*, a task expressed as a long-term goal of the GAN research project. To that end, we train ResNet-50 classifiers using either purely BigGAN images or mixtures of ImageNet and BigGAN images, and test on the ImageNet validation set. Our preliminary results suggest both a measured view of state-of-the-art GAN quality and highlight limitations of current metrics. Using only BigGAN images, we find that Top-1 and Top-5 error increased by 120% and 384%, respectively, and furthermore, adding more BigGAN data to the ImageNet training set at best only marginally improves classifier performance. Finally, we find that neither Inception Score, nor FID, nor combinations thereof are predictive of classification accuracy. These results suggest that as GANs are beginning to be deployed in downstream tasks, we should create metrics that better measure downstream task performance. We propose classification performance as one such metric that, in addition to assessing per-class sample quality, is more suited to such downstream tasks.

## 1 Introduction

Recent years have witnessed a marked improvement in sample quality in Deep Generative Models. One model class in particular, Generative Adversarial Networks (Goodfellow et al., 2014), has begun to generate nearly photorealistic images. While applications of adversarial training have found their way into image translation (Zhu et al., 2017) and style transfer (Chan et al., 2018), a typically discussed goal for such models, and in particular conditional ones, is data augmentation. Such models have enjoyed limited success in these tasks thus far for large-scale datasets such as ImageNet, likely because existing models did not generate sufficiently high-quality samples. Recently, however, BigGANs (Brock et al., 2018) have generated photorealistic images of ImageNet data up to 512×512 resolution, and moreover, achieve Inception Scores and Frechet Inception Distances similar to the dataset on which they were trained. Such results suggest, though do not prove, that BigGANs are indeed capturing the data distribution. If this were true, then it seems plausible that these samples can be used in downstream tasks, especially in situations in which limited labelled data are available.

In this work, we test the rather simple hypothesis that BigGANs are indeed useful for *data augmentation*, or more drastically, *data replacement* of the original data distribution. To that end, we use BigGANs for two simple experiments. First, we train ImageNet classifiers, replacing the original training set with one produced by BigGAN. Second, we augment the original ImageNet training set with samples from BigGAN. Our working hypothesis is that if BigGANs were indeed capturing the data distribution, we could use those samples, instead of or in addition to the original training set, to improve performance on classification. That it does not – on replacement, Top-5 classification

Table 1: ResNet-50 Results for *replacement* experiments for BigGAN.

| Training Set | Truncation | Resolution | Top-5 Error | Top-1 Error | Inception Score | FID-50K |
|---|---|---|---|---|---|---|
| Real | - | 256×256 | 7.03% | 25.99% | $331.83 \pm 5.0$ | 2.47 |
| BigGAN-deep | 0.20 | 256×256 | 86.76% | 94.89% | $\mathbf{339.06 \pm 3.14}$ | 20.75 |
| BigGAN-deep | 0.42 | 256×256 | 71.32% | 86.70% | $324.62 \pm 3.29$ | 15.93 |
| BigGAN-deep | 0.50 | 256×256 | 67.12% | 84.34% | $316.31 \pm 3.70$ | 14.37 |
| BigGAN-deep | 0.60 | 256×256 | 54.99% | 74.49% | $299.51 \pm 3.20$ | 12.41 |
| BigGAN-deep | 0.80 | 256×256 | 43.32% | 67.12% | $258.72 \pm 2.86$ | 9.24 |
| BigGAN-deep | 1.00 | 256×256 | 37.03% | 60.93% | $214.64 \pm 2.01$ | **7.42** |
| BigGAN-deep | 1.50 | 256×256 | **34.08**% | **57.35**% | $109.39 \pm 1.56$ | 11.78 |
| BigGAN-deep | 2.00 | 256×256 | 35.63% | 59.02% | $49.54 \pm 0.98$ | 28.67 |

error increased by 384% compared to the original training set; and on augmentation, classification performance improves only marginally while dramatically increasing training time – suggests that naively augmenting the dataset with BigGAN samples is of limited utility and more work is required for BigGANs to be used in downstream tasks.

Though a negative result, a more positive byproduct of the work is the introduction of a new metric that can better identify issues with GAN and other generative models. In particular, training a classifier allows us to identify, for conditional generative models, which classes are particularly poor, either due to low quality samples or underrepresentation of dataset diversity.

## 2 EXPERIMENTS

### 2.1 SETUP

Our experiments are rather simple: we use BigGAN-deep (further denoted as BigGAN) models to either *replace* or *augment* the ImageNet training set, train an image classifier, and compare performance on the ImageNet validation set. In the *data replacement* experiments, we replace the ImageNet training set with one from BigGAN-deep, and each example from the original training set is replaced with a model sample from the same class. In the *augmentation* experiments, we add to the ImageNet training set, 25%, 50%, or 100% more data from BigGAN. Moreover, since the truncation trick – which resamples dimensions that are outside the mean of the distribution – seems to trade off quality for diversity, we perform experiments for a sweep of truncation parameters: 0.2, 0.42, 0.5, 0.8, 1.0, 1.5, and 2.0.[1] In addition, we compare performance on replacement and augmentation to two traditional GAN metrics: Inception Score (Salimans et al., 2016) and Frechet Inception Distance (FID) (Heusel et al., 2017), as these metrics are the current gold standard for GAN comparison. Both rely on a feature space from a classifier trained on ImageNet, suggesting that if metrics are useful at predicting performance on a downstream task, it would indeed be this one.

We used ResNet-50 (He et al., 2016) classifier for our models, with single-crop evaluation. The classifier is trained for 90 epochs using TensorFlow's momentum optimizer, a learning rate schedule linearly increasing from 0.0 to 0.4 for the first 5 epochs, and decreased by a factor of 10 at epochs 30, 60, and 80. It mirrors the 8,192 batch setup of Goyal et al. (2017) with gradual warmup.

### 2.2 RESULTS

Table 1 shows the performance of classifiers trained on BigGAN datasets compared to the real dataset. At every truncation level, ResNet-50 classifiers trained on BigGAN samples generalize substantially worse to real images than the classifier trained on real data. To better understand why this has occurred, we broke down the performance by class for the best-performing truncation level: 1.5. As shown in the left pane of Figure 1, nearly every class suffers a drop in performance

---

[1]Dimensions of the noise vector $z$ whose value are greater outside the range of $-2\tau$ to $2\tau$ ($\tau$ is the truncation parameter) are resampled. Lower values of $\tau$ lead to less diverse datasets.

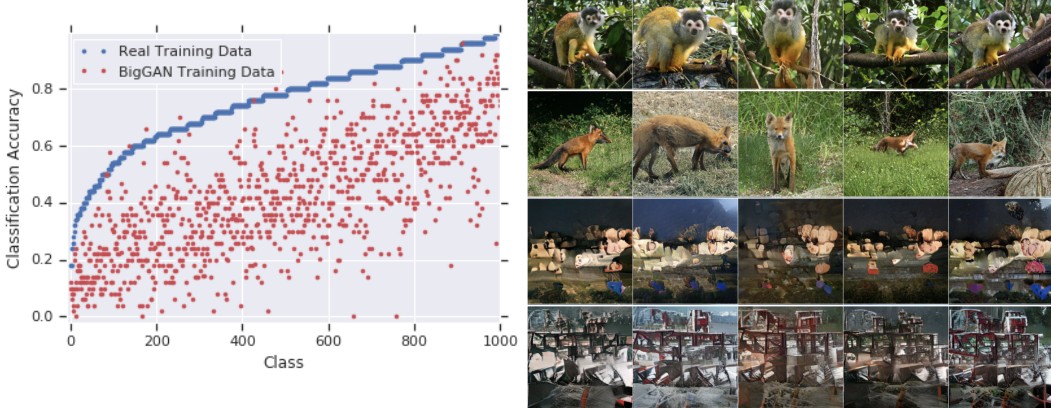

Figure 1: Left: comparison of per-class accuracy of original (blue) vs. BigGAN-deep (red) training data at 1.5 truncation level. Right: the top two rows are the BigGAN-deep samples from classes – squirrel monkey, and red fox – that achieved the best test set performance relative to original dataset. The bottom two rows are those from classes – balloon and paddlewheel – which achieved the worst.

compared to the original dataset. Performance on six classes – partridge, red fox, jaguar/panther, squirrel monkey, African elephant, and strawberry – did improve over the original dataset, though the improvement for those classes was marginal. The right pane of Figure 1 shows the two best and two worst performing categories, as measured by the difference in classification performance. Notably, for the two worst performing categories and two others – balloon, paddlewheel, pencil sharpener, and spatula – classification accuracy was 0% on the validation set. That said, at the best truncation levels, Top-5 Error is roughly 35%, suggesting that BigGANs are learning nontrivial distributions.

Given the performance of BigGAN in the replacement experiments, one should not necessarily expect improved classifier accuracy by augmenting ImageNet training set with BigGAN samples. Figure 2 illustrates the performance of the classifiers when we increase the amount of BigGAN training data. Perhaps somewhat surprisingly, BigGAN models that sample from lower truncation values, and have lower sample diversity, are able to perform better on data augmentation compared to those models that performed the best on the data replacement experiment. In fact, for some of the lowest truncation values, one found modest improvement in classification performance: roughly 3% improvement relative on Top-1 Error (but at the cost of 1.5 times the amount of training time).

Finally, Inception Score and FID had very little correlation with performance on either the replacement or augmentation experiments, suggesting that alternative metrics will be needed when we turn our attention to downstream tasks. For our replacement experiments, the correlation coefficient between Top-1 error and FID is 0.16, and Inception Score 0.86, the latter result incorrectly suggesting that improved Inception Score is highly correlated with increased error. Moreover, the best-performing methods have rather poor Inception Score and FIDs. That models that perform poorly on Inception Score and Frechet Inception Distance also perform poorly on classification is no surprise; that models that perform well on Inception Scores and FID perform poorly on classification suggests that alternative metrics are needed. One can easily diagnose the issue with Inception Score: as Barratt & Sharma (2018) noted, Inception Score does not account for intra-class diversity, and a training set with little intra-class diversity may make the classifier fail to generalize to a more diverse test set. FID should better account for this lack of diversity at least grossly, as the metric, calculated as $FID(P_x, P_y) = \|\mu_x - \mu_y\|^2 + tr(\Sigma_x + \Sigma_y - 2(\Sigma_x \Sigma_y)^{1/2})$, compares the covariance matrices of the data and model distribution. By comparison, per-class classification error offers a finer measure of model performance, as it provides us a per-class metric to identify which classes have better or worse performance. While in theory one could calculate a per-class FID, FID is known to suffer from high bias (Bińkowski et al., 2018) for low number of samples, likely making the per-class estimates unreliable. [2]

---

[2]Bińkowski et al. (2018) proposed Kernel Inception Distance, an unbiased alternative to FID, but this metric suffers from variance too large to be reliable when using the number of per-class samples in the ImageNet training set (roughly 1,000 per class), much less when using the 50 in the validation set.

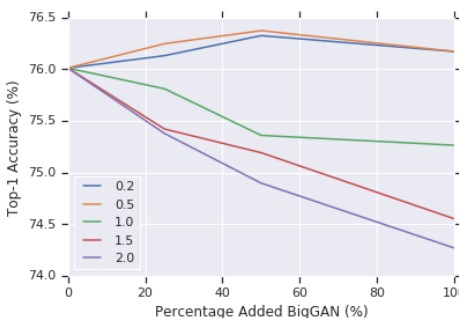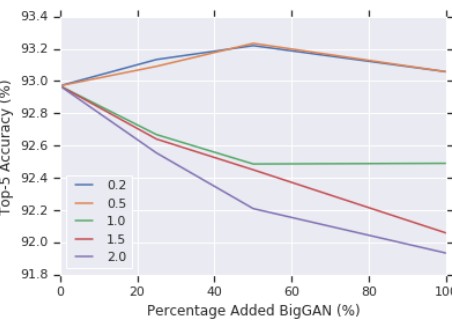

Figure 2: Top-1 (left) and Top-5 (Right) accuracy as training data is augmented by $x\%$ examples from BigGAN-deep for different truncation levels. Lower truncation generates datasets with less sample diversity.

The results on augmentation highlight different desiderata for samples that are added to the dataset rather than replaced. Clearly, the samples added should be sufficiently different from the data to allow the classifier to better generalize, and yet, poorer sample quality may lead to poorer generalization compared to the original dataset. This may be the reason why extending the dataset with samples generated from a lower truncation value noise – which are higher-quality, but less diverse – perform better on augmentation than replacement. Furthermore, this may also explain why Inception Score, Frechet Inception Distance, and data replacement classification error are not predictive of data augmentation classification performance.

## 3 RELATED WORK

This work encompasses two lines of work: GANs for data augmentation and improved evaluation metrics for GANs. For data augmentation, Antoniou et al. (2017) proposed an image-conditioned model for augmentation, and found improved results on smaller datasets. Frid-Adar et al. (2018) used a GAN to generate synthetic training data of size $64 \times 64 \times 1$ of images of liver lesions. For evaluation metrics, Theis et al. (2016) noted there is difficulty in designing evaluation metrics that will illustrate the general performance in the model. Despite this finding, those interested in measuring the quality of implicit generative models have proposed practical metrics to compare sample quality from different models: which have led to introduction of Inception Score and FID. Lopez-Paz & Oquab (2016) recommends the use classifier two-sample tests to test GAN samples as a metric. Other measures attempt to determine other properties of generative models. Lucic et al. (2018) constructs synthetic datasets for which precision and recall can be computed approximately and compares Inception Score and FID to changes in precision and recall. Geometry Score (Khrulkov & Oseledets, 2018) constructs approximate manifolds from data and samples, and uses them for GAN samples to determine whether there was mode collapse. Arora & Zhang (2017) attempt to determine the support size of GANs by using a Birthday Paradox test, though it requires a human to identify two nearly-identical samples.

## 4 CONCLUSION

In this work, we investigated to what extent BigGAN, the state-of-the-art GAN on ImageNet, captures the data distribution, and to what extent those samples can be used for data augmentation. Our results demonstrate that despite excellent scores on traditional GAN metrics such as Inception Score and Frechet Inception Distance, current state-of-the-art GAN models do not capture the distribution for large-scale datasets such as ImageNet. Moreover, we found only a modest improvement in classifier performance when the training set was augmented with BigGAN samples. Finally, through classifier metrics outlined in the work, we can identify on which classes BigGAN performed well, and on which ones researchers should focus their future efforts.

An open question in this work is how to create metrics predictive of performance on downstream tasks. Even for the classifier metric, results on data replacement did not necessarily correlate with those on data augmentation. Better evaluation metrics will help us understand to what extent GANs, or any other Deep Generative Models, can be used for downstream tasks.

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
