# OpenReview forum: "Seeing is Not Necessarily Believing: Limitations of BigGANs for Data Augmentation"
_ICLR.cc/2019/Workshop/LLD — LLD 2019_

### Official Review · AnonReviewer2 · 2019-04-08
**Detailed exploration of GAN in a different setting**

**Rating:** 4
**Confidence:** 3

**Review:**

This paper clearly sets out a hypothesis, method to test the hypothesis, and the presents the core (negative) result "BigGANs cannot be currently used for data augmentation and more work is required for it to be
used in downstream tasks". This work overall is clear and direct, and I enjoyed reading and reviewing it. I will focus the remainder of the review on suggestions, questions and ideas for perhaps pushing this work in some other directions.

The experiments here are quite detailed, though the nature of the work opens a whole new series of questions that I hope the authors continue exploring. Rather than only using the model for data augmentation, the images / representations for the classifier could also be augmented using discriminator features extracted learned by the learned-and-frozen GAN - this could perhaps still be an improvement even if the direct images / dataset augmentation isn't sufficient to improve classification. These types of "bootstrapping" of the GAN with discriminator features can be seen in papers such as "Denoising Feature Matching", though used in a different setting https://openreview.net/forum?id=S1X7nhsxl .

One big question I have is in regard to the use of GAN (trained on very large datasets, potentially) as a regularization technique for much smaller datasets than ImageNet. This setting might allow the use of much larger models on small data, due to better regularization - though ImageNet has been (slightly) overfit these days, it is still enormous compared to many practical "in-the-wild" datasets and seems a particularly hard test for this data augmentation setup. Out-of-domain generalization is potentially another area to explore with practical application.

---

### Official Review · AnonReviewer1 · 2019-04-09
**Good systematic study of the inadequacy of GAN for data-augmentation**

**Rating:** 4
**Confidence:** 2

**Review:**

The authors train ResNet-50 networks on a mixture of ImageNet data and BigGAN samples and show that replacing ImageNet data with BigGAN samples leads to a decrease in performance.

Positives:
- This "data-replacement" experiment is very natural to make, therefore I am glad it was done.
- The metric for identifying BigGAN failures per class is also an interesting byproduct.

Remarks:
a) "per-class FID" is said to be likely "making the per-class estimates unreliable" due to high variance. I would still like to see what the per-class FID gives and how the best and worst-performing classes compare with the one found by this method. Indeed, even if per-class FID could perform worse, it has the great benefit of not necessitating to train new Resnet-50 networks.
b) I would be interested to see some samples generated with the best truncation for replacement, and for addition, in order to have a better idea of what kind of "diversity" we are talking of (this could replace one of the two subplots of Fig. 2 since top1/top5 follow the same trends).

Overall, I think this paper and its experiments is a nice contribution to the workshop.

---

### Decision · Program_Chairs · 2019-04-09
**Acceptance Decision**

Accept